# A Tailored Antithrombotic Approach for Patients with Atrial Fibrillation Presenting with Acute Coronary Syndrome and/or Undergoing PCI: A Case Series

**DOI:** 10.3390/jcm11144089

**Published:** 2022-07-14

**Authors:** Simona Giubilato, Fabiana Lucà, Andrea Pozzi, Giorgio Caretta, Stefano Cornara, Anna Pilleri, Concetta Di Nora, Francesco Amico, Irene Di Matteo, Silvia Favilli, Roberta Rossini, Carmine Riccio, Furio Colivicchi, Michele Massimo Gulizia

**Affiliations:** 1Cardiology Department, Cannizzaro Hospital, 95126 Catania, Italy; famico64@gmail.com; 2Cardiology Department, Grande Ospedale Metropolitano, AO Bianchi Melacrino Morelli, 89129 Reggio Calabria, Italy; fabiana.luca92@gmail.com; 3Cardiology Department, Papa Giovanni XXIII Hospital, 24127 Bergamo, Italy; andreawellsvabg@gmail.com; 4Sant’Andrea Hospital, ASL 5 Regione Liguria, 19124 La Spezia, Italy; giorgio.caretta@gmail.com; 5Arrhytmia Unit, Division of Cardiology, Ospedale San Paolo, Azienda Sanitaria Locale 2, 17100 Savona, Italy; stefano.cornara@gmail.com; 6Cardiology Unit, Brotzu Hospital, 09121 Cagliari, Italy; annapilleri@aob.it; 7Department of Cardiothoracic Science, Azienda Sanitaria Universitaria Integrata di Udine, 33100 Udine, Italy; concetta.dinora@gmail.com; 8Cardiology, ASST Grande Ospedale Metropolitano Niguarda, 20162 Milan, Italy; irene.dimatteo@ospedaleniguarda.it; 9Department of Pediatric Cardiology, Meyer Hospital, 50139 Florence, Italy; silvia.favilli@meyer.it; 10Cardiology Unit, Ospedale Santa Croce e Carle, 12100 Cuneo, Italy; roberta.rossini2@gmail.com; 11Cardiovascular Department, Sant’Anna e San Sebastiano Hospital, 81100 Caserta, Italy; carmine.riccio8@icloud.com; 12Clinical and Rehabilitation Cardiology Department, San Filippo Neri Hospital, ASL Roma 1, 00100 Roma, Italy; furio.colivicchi@gmail.com; 13Cardiology Department, Garibaldi Nesima Hospital, 95122 Catania, Italy; michele.gulizia60@gmail.com; 14Fondazione per il Tuo Cuore-Heart Care Foundation, 50121 Firenze, Italy

**Keywords:** atrial fibrillation, acute coronary syndrome, DOAC

## Abstract

The combination of oral anticoagulants (OAC) and dual antiplatelet therapy (DAPT) is the mainstay for the treatment of patients with atrial fibrillation (AF) presenting with acute coronary syndrome (ACS) and/or undergoing PCI. However, this treatment leads to a significant increase in risk of bleeding. In most cases, according to the most recent guidelines, triple antithrombotic therapy (TAT) consisting of OAC and DAPT, typically aspirin and clopidogrel, should be limited to one week after ACS and/or PCI (default strategy). On the other hand, in patients with a high ischemic risk (i.e., stent thrombosis) and without increased risk of bleeding, TAT should be continued for up to one month. Direct oral anticoagulants (DOAC) in triple or dual antithrombotic therapy (OAC and P2Y12 inhibitor) should be favored over vitamin K antagonists (VKA) because of their favorable risk/benefit profile. The choice of the duration of TAT (one week or one month) depends on a case-by-case evaluation of a whole series of hemorrhagic or ischemic risk factors for each patient. Likewise, the specific DOAC treatment should be selected according to the clinical characteristics of each patient. We propose a series of paradigmatic clinical cases to illustrate the decision-making work-up in clinical practice.

## 1. Introduction

Atrial fibrillation (AF) and coronary artery disease (CAD) frequently coexist in the same patient [1]. It has been reported that CAD affects more than 20% of patients with AF [2]. Conversely, AF occurs in about 15% of patients with acute coronary syndrome (ACS), and one-third of those are newly diagnosed [3]. Remarkably, when AF and ACS are associated, or if percutaneous coronary intervention (PCI) is performed in a patient with AF, the antithrombotic strategy is more challenging. Indeed, although the combination of oral anticoagulation (OAC) and dual antiplatelet therapy (DAPT) is required to reduce the risk of both thromboembolic and ischemic events, this treatment leads to a significant increase in bleeding risk, up to four times higher compared with OAC alone [3].

A short period of triple antithrombotic therapy (TAT), consisting of OAC and DAPT, typically aspirin and clopidogrel, is recommended [4,5,6,7]. However, considering the increased mortality related to major bleeding [8], the optimal antithrombotic regimen should be carefully evaluated in order to reduce the risk of bleeding.

Therefore, in this setting, the assessment of both ischemic and bleeding risk in each patient is needed to personalize a specific antithrombotic/antiplatelet regimen in terms of type, dosing, and duration in order to achieve a net clinical benefit. We propose paradigmatic clinical cases to illustrate decision-making workups in clinical practice.

### 1.1. Patient 1

A 69 year old man with a history of diabetes, hypertension, and non-valvular atrial fibrillation (NVAF) in treatment with vitamin K antagonists (VKA) was admitted with acute anterolateral myocardial infarction with ST elevation (STEMI). Pre-treatment with 300 mg of aspirin and 300 mg of clopidogrel was administered. Invasive coronary angiography (ICA) showed 90% stenosis of the first proximal obtuse marginal artery and acute occlusion of the left anterior descending artery (LAD) (Figure 1). A two-stent strategy was necessary in order to treat the culprit lesion with the implantation of two polymer-free drug-eluting stents (DES) (Figure 2A). Blood tests showed 13.5 g/dL of haemoglobin, 167,000/mL of platelets, 0.7 mg/dL of creatinine (creatinine clearance (CrCl) was 58 mL/min), INR of 2.6, normal liver enzyme levels, and troponin I hs 62 ng/mL. On day 4, the patient received staged PCI and implantation of a DES in the proximal obtuse marginal artery (Figure 2B). He had a CHA2DS2-VASc score of 4 and a HAS-BLED score of 2. OAC with 150 mg b.i.d of dabigatran was started on a background of low-dose aspirin and clopidogrel. The patient was discharged on day seven on triple therapy considering his high ischaemic risk due to clinical presentation (ACS and, in particular, STEMI) and other anatomical/procedural characteristics (bifurcation with two stents implanted, three lesions and stents implanted, total stent length > 60 mm). The treatment strategy was TAT for one month, followed by dabigatran and clopidogrel (DAT) for 12 months. He did not present recurrent ischemic events at the one-year follow-up.

### 1.2. Patient 2

A 68 year old female presented with stable angina Canadian Cardiovascular Society (CCS) grading III and positive stress-echocardiography was referred to ICA. She had not well-treated hypertension, hypercholesterolemia, and chronic kidney disease (CKD). She also reported a history of palpitations and, 3 years before admission, an episode of melena. Laboratory data showed 10 g/dL of haemoglobin, 155.000/mL of platelets, 1.2 mg/dL of creatinine (ClCr was 35 mL/min), and normal liver enzyme levels. The transthoracic echocardiography (TTE) demonstrated left ventricular hypertrophy (LVH) in the presence of normal ejection fraction (EF), mild left atrial (LA) dilation, and mild mitral regurgitation (MR). She underwent ICA, which revealed a significant lesion of a large diagonal branch treated with PCI and implantation of one DES (Figure 3).

The patient was treated with aspirin and a loading dose of 600 mg of clopidogrel. On the same day, she developed palpitations, and the ECG showed AF with a high ventricular rate (Figure 4). A pharmacological cardioversion with intravenous amiodarone and subcutaneous enoxaparin for thromboembolic prevention was performed. After a few hours, there was a recovery of sinus rhythm (SR) but episodes of paroxysmal AF were observed in the following days. Both the CHA2DS2-VASc and HAS-BLED scores were 4. Aspirin was discontinued, and the patient was discharged with 75 mg/day of clopidogrel and 15 mg/day of rivaroxaban. At the six-month follow-up, there were no ischaemic and bleeding events; clopidogrel was discontinued, whereas the oral anticoagulant was maintained.

### 1.3. Patient 3

A 64 year old man with hypertension, hypercholesterolemia, AF, and moderate CKD was admitted to our department with chest pain. Eighteen months prior, he had non-STEMI (NSTEMI) treated with PCI and implantation of a DES on LAD, ramus intermedius (RI), and right coronary artery (RCA). He had no liver disease. Medical treatment included warfarin, 100 mg of aspirin daily, 5 mg of bisoprolol daily, 5 mg of ramipril b.i.d., and 40 mg of atorvastatin daily. Physical examination revealed irregular pulse and blood pressure (BP) of 130/60 mmHg with no signs of congestive heart failure (CHF). He had a respiratory rate of 16/min with a peripheral O_2_ saturation of 97% upon examination. Auscultation did not reveal any abnormal breathing sounds, rales, or rhonchi. The cardiac examination revealed an irregular rhythm with no murmurs, gallops, or rubs. There was no peripheral edema. His laboratory investigations were as follows: 13 g/dL of haemoglobin, INR 1.8, 1.4 mg/dL of serum creatinine (CrCl: was 52 mL/min), and I hs 600 ng/mL troponin (normal < 0.04 ng/mL). Twelve-lead electrocardiogram upon admission showed AF at 68 b.p.m. without any specific ST-T segment changes. The LVEF was 48%. Both the CHA2DS2-VASc and HAS-BLED scores were 2. The patient underwent a transradial ICA, which showed the patency of previously implanted stents without any coronary stenosis (Figure 5).

Considering the presence of ACS and atrial fibrillation, the patient was switched from warfarin plus aspirin to 5 mg of apixaban b.i.d. plus clopidogrel. During the following days, the patient remained asymptomatic and was discharged on the fourth day in DAT for at least 6 months. No ischemic recurrences were reported at the 6-month follow-up. The patient reported minor bleeding (bruising and bleeding gums). Therefore, we decided to drop clopidogrel and continue with 5 mg of apixaban b.i.d. only.

### 1.4. Patient 4

A 67 year old woman with prior myocardial infarction (MI) and hypertension was admitted to our coronary care unit (CCU) with a diagnosis of STEMI and acute pulmonary edema. She had NVAF treated with VKA and a history of hypersensitivity to aspirin.

TTE showed left ventricular dilatation with a severely reduced left ventricular ejection fraction (LVEF 30%) and functional moderate mitral regurgitation. The twelve-lead electrocardiogram (ECG) revealed AF and a left bundle branch block. The blood tests showed normal renal and liver function and no anemia. The ICA revealed severe multi-vessel disease with critical stenosis at the left main (LM), extending distally to the proximal LAD, and significant in-stent restenosis of the RCA. The patient declined coronary artery bypass graft surgery (CABG); hence, she underwent a successful re-PCI of RCA, and on day 6, she had staged PCI with DES implantation on the LM to the proximal LAD. Intravascular ultrasound (IVUS) was performed for optimizing the left main (LM)-LAD stent. She had a CHA2DS2-VASc score of 5 and a HAS-BLED score of 2.

Given her high clinical and procedural ischaemic risk (ACS and multiple vessels disease with LM involvement) and the unfeasibility of treatment with TAT due to hypersensitivity to aspirin, the patient was discharged with 150 mg of dabigatran b.i.d. and 90 mg of ticagrelor b.i.d, a more effective P2Y12 inhibitor than clopidogrel, for 12 months. She remains well at the 12-month follow-up.

### 1.5. Patient 5

An 85 year old frail man with NVAF receiving treatment with 30 mg of edoxaban was admitted to our CCU for inferior STEMI. He underwent transradial primary PCI of the RCA with the placement of a DES. A dose loading of aspirin (250 mg i.v.) and clopidogrel (600 mg) was administered. Laboratory data showed 7.8 g/dL of haemoglobin, 80.000/mL of platelets, 1.8 mg/dL of creatinine (ClCr 30 mL/min), and normal liver enzyme levels. Stratification of both the risk of stroke and bleeding was performed (CHA2DS2-VASc score 4 and HAS-BLED score 2).

According to the ARC-HBR criteria and PRECISE-DAPT score, the patient had a high bleeding risk (HBR). Accordingly, TAT with edoxaban, aspirin, and clopidogrel was limited to the periprocedural phase, and the patient was discharged after five days of hospitalization on DAT (edoxaban plus clopidogrel) for up to 6 months. He was closely followed up thereafter. At the monthly follow up, the hemoglobin values were stable or slightly rising. The patient remained asymptomatic. Therefore, clopidogrel was discontinued after 6 months.

## 2. Discussion

We report five cases of patients with AF presenting with ACS and/or treated by PCI to show the decision-making work-up in clinical practice regarding the choice of antithrombotic regimen. In this scenario, DOAC should be preferred over VKA because of their favorable risk/benefit profile, as recommended by the current guidelines. The choice of the optimal antithrombotic therapy (TAT or DAT) and the duration of TAT (one week or one month) depends on a careful evaluation of the individual patient’s hemorrhagic and ischemic risk factors, as well as evaluation of the coronary anatomy profile and procedural complexity in order to identify patients who might benefit from prolonged TAT and those who might have an excessive risk of bleeding. The choice of specific DOAC and dosage represent the most important challenge in these patients, and should be based on clinical characteristics and hemorrhagic and ischemic risk (previous OAC therapy, frailty, renal function, presence of criteria for dose reduction, etc.). The different factors to consider when determining the optimal antithrombotic regimen for individual patients are summarized in Figure 6. Moreover, in Figure 7, we suggest a practical algorithm for the choice of antithrombotic treatment in patients with atrial fibrillation presenting with acute coronary syndrome and/or undergoing PCI.

In patients with ACS and NVAF, the safety and efficacy of DAT, consisting of a DOAC with a P2Y12 inhibitor, usually clopidogrel, have been specifically addressed in different randomized trials [9,10,11,12], reporting a significantly lower bleeding risk of DAT compared with TAT without increasing the major adverse cardiac events (MACE).

However, these trials were drawn to assess the risk of bleeding, while they were underpowered in terms of the sample size and follow-up length in order to investigate the benefit on ischemic risk (i.e., stent thrombosis) [9,10,11,12]. Another major limitation of these trials was the mixture of stable CAD and ACS patients, as approximately half of the patients had ACS, whereas less than 15% had STEMI.

Several meta-analyses investigated the risk of an ischemic event in the aforementioned trials, with different results. While two meta-analyses [13,14] reported a reduced risk of bleeding without a significantly increased risk of coronary thrombosis with DAT compared with TAT, others found a small but statistically significant increase in the risk of coronary events, such as stent thrombosis and MI [4,5,15,16].

In patients with a TAT regime, the use of the newer and more powerful P2Y12-receptor inhibitors, prasugrel and ticagrelor, has been discouraged based on safety concerns [6,17], as a greater risk of major bleeding compared with clopidogrel has been reported [18,19,20,21,22]. In the RE-DUAL PCI trial [12], ticagrelor was combined with dabigatran in 12% of patients, whereas this combination was less frequent in PIONEER-AF [9], AUGUSTUS [10], and ENTRUSTAF PCI [14].

Recent guidelines and consensus documents [6,7,17,23] recommend the use of DOAC over VKA in DOAC eligible patients at the recommended dose for stroke prevention (Class I) [6]. However, according to the results of PIONEER-AF PCI, a lower dose of rivaroxaban (i.e., 15 mg once daily) could be considered when used in combination with aspirin and/or clopidogrel (class IIb) [6]. Conversely, a reduced dose of apixaban (2.5 mg b.i.d) and edoxaban (30 mg o.d.) should not be used in the absence of drug-specific criteria for dose reduction.

DAT with DOAC and single antiplatelet therapy, up to 12 months after one week of TAT, should be used as the default strategy according to the 2020 ESC guidelines [6]. In patients at high risk of bleeding, DAT should be shortened to 6 months, whereas in patients where the ischaemic risk (based on clinical, anatomical, or procedural characteristics) outweighs the bleeding risk, TAT should be continued for up to 1 month.

We report five cases of patients with AF presenting with ACS and/or treated by PCI to show the decision-making work-up in clinical practice regarding the choice of the antithrombotic regimen. Our first case showed a patient with NVAF and ACS who underwent primary PCI with high-risk clinical and angiographic features for ischemic coronary outcomes who was at low risk of bleeding. According to current guidelines, in this case, we prescribed TAT with a DOAC and DAPT (aspirin plus clopidogrel) for 1 month following PCI. A loading dose of 300 mg of clopidogrel seems to be a reasonable choice in patients treated with VKA and with an unknown INR value, in order to reduce the risk of bleeding. Moreover, following the guideline recommendations that support the use of DOACs over VKA, as a combination therapy with antiplatelets, we switched from VKA to dabigatran. The 150 mg dose of dabigatran was chosen due to the patient’s clinical characteristics (CHA_2_DS_2_-VASc score: 4; HAS-BLED score: 2; creatinine clearance > 50 mL/min). According to the current ESC guidelines, after an appropriate TAT treatment, this patient needed to be treated with dabigatran and clopidogrel for up to 12 months, followed by OAC with dabigatran as the chronic treatment [6]. In this regard, the recent AFIRE trial demonstrated the safety in transitioning to DOAC monotherapy without any antiplatelet agent beyond 1 year after cardiac revascularization in AF patients [24].

In the second scenario, we described a patient with high hemorrhagic and lower ischemic risk. The patient had a very high bleeding risk, not only due to her HAS-BLED score > 3, but also because she was female; moreover, she had anemia and a previous history of bleeding. On the other hand, she underwent a simple one vessel, single stent PCI with implantation of a new generation DES for stable CAD.

Although there is no clear evidence that the periprocedural onset of AF in patients undergoing PCI is associated with a comparable risk of pre-existent AF, the current guidelines [6] recommend OAC according to the individual’s thromboembolic risk (our patient CHA_2_DS_2_-VASc score: 4).

Considering all of the aforementioned characteristics, our patient was discharged with short peri-procedural TAT followed by a short DAT (for up 6 months) to balance the hemorrhagic and ischemic risk. According to her high bleeding risk (previous hemorrhage), age, and reduced renal function (eGFR 35 mL/min), we prescribed a reduced dose of rivaroxaban.

The third case described a patient with NSTEMI-ACS managed medically. According to registries, this challenging population accounts for almost one-third of the ACS population.

In this regard, AUGUSTUS was the only DOAC-trial that included patients with medically managed ACS (approximately 23% of enrolled patients) [25]. The risk of bleeding with apixaban was 56% lower compared with VKA. Importantly, the risk of bleeding was 49% higher with aspirin compared with a placebo, without a significant difference in death and ischemic events. The results of this analysis support a DAT consisting of apixaban with clopidogrel for at least the first 6 months, which is the therapeutic option we used in our case. Moreover, differently from the other studies, the factorial design of AUGUSTUS demonstrated the greater safety of DAT with DOAC plus P2Y12 compared with the conventional TAT with VKA P2Y12, and aspirin is attributable to both the use of DOAC versus VKA and early aspirin discontinuation.

The fourth patient represents a case to ideally be a candidate for TAT, in which TAT is not applicable for the history of aspirin hypersensitivity. In this post-ACS patient at high risk of coronary thrombotic and low risk of bleeding, a DOAC-based DAT with a more potent P2Y12 inhibitor, ticagrelor, could represent a reasonable treatment option (Class IIb in current guidelines) [23].

The REDUAL-PCI is the largest trial that assessed combination therapy with ticagrelor and DOAC. A prespecified subgroup analysis from the RE-DUAL PCI trial showed fewer bleeding events, even in patients treated with dabigatran, irrespective of the dose, plus ticagrelor, compared with those treated with warfarin TAT [26]. According to these results, we prescribed dabigatran as combination therapy with ticagrelor.

The last case described an elderly patient with STEMI at high risk of bleeding who was already being treated with a correctly reduced dose of DOAC.

In this setting, we used the ESC guideline’s recommendation therapy consisting of TAT with clopidogrel during the in-hospital length of stay, followed by DAT with clopidogrel for a maximum of six months.

Other strategies to avoid bleeding complications in all of the patients were low-dose aspirin (≤100 mg) usage and the routine use of proton pump inhibitors for gastric protection.

## 3. Conclusions

The management of patients with AF presenting with acute ACS and/or undergoing PCI who need a combination of anticoagulant and antiplatelet therapy remains a common and controversial issue in clinical practice, and there is no one-size-fits-all antithrombotic treatment for these patients. We sought to describe a practical approach to implementing the current guidelines and evidence-based data into real world clinical practice by reporting a series of paradigmatic clinical cases. As demonstrated in this case series, a patient-tailored approach is crucial for the management of antithrombotic therapy in the setting of AF and ACS or PCI.

## Figures and Tables

**Figure 1 jcm-11-04089-f001:**
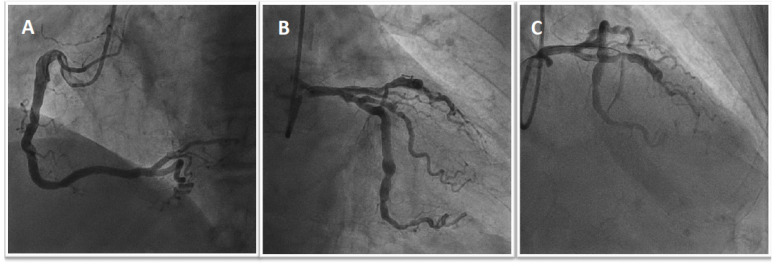
Urgent coronary angiography shows atherosclerosis of the right coronary artery without significant stenosis (**A**), severe stenosis of the first proximal obtuse marginal artery (**B**), and acute occlusion of the mid-LAD with the presence of thrombus (**C**).

**Figure 2 jcm-11-04089-f002:**
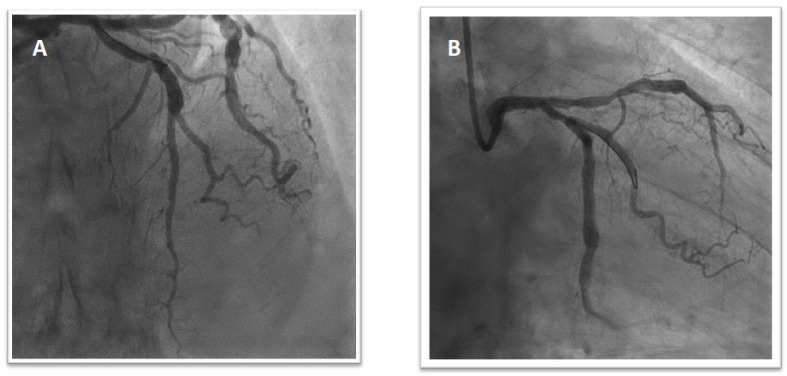
Final result after the first intervention with the implantation of two polymer-free DES on the mid-LAD and second diagonal branch (T stent technique; 2.75 × 24 mm and 2.25 × 14 mm) (**A**). Final result after the second intervention with implantation of a DES in the proximal obtuse marginal artery (2.5 × 25 mm) (**B**).

**Figure 3 jcm-11-04089-f003:**
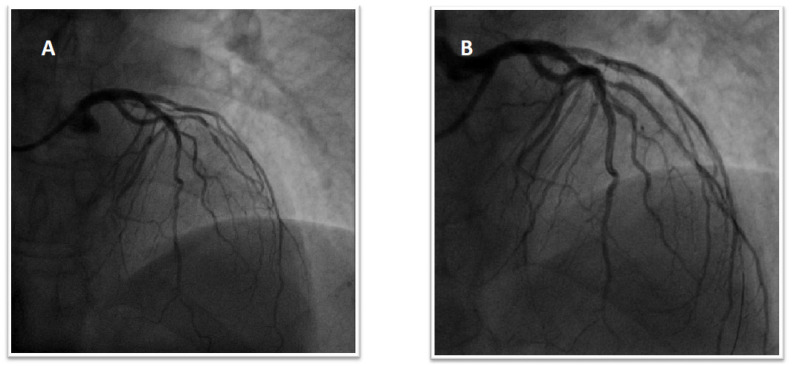
Angiogram showing critical stenosis of a large diagonal branch (**A**) treated with PCI and implantation of one DES (2.75 × 18 mm) (**B**).

**Figure 4 jcm-11-04089-f004:**
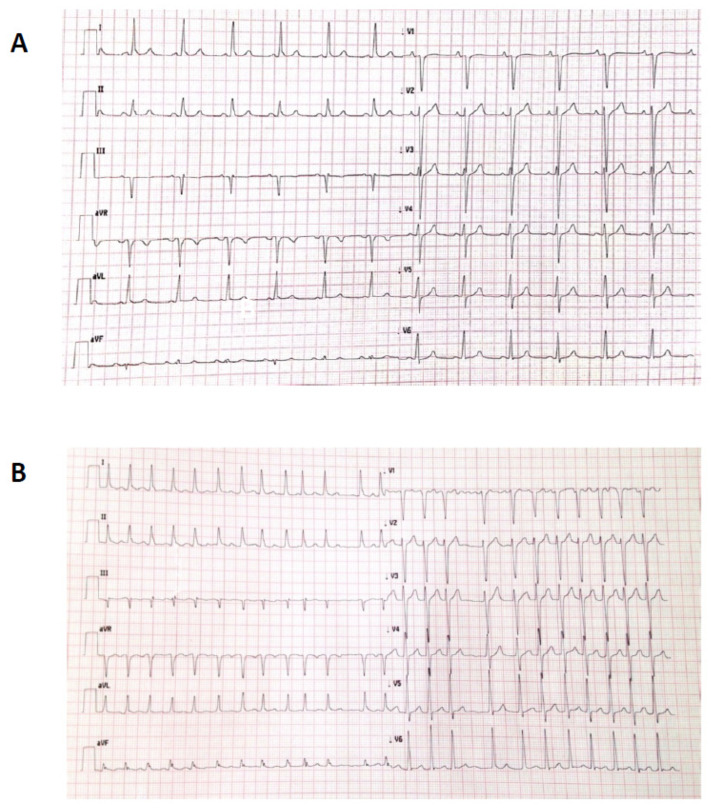
(**A**). Patient’s electrocardiogram at presentation. (**B**). Patient’s electrocardiogram after PCI showing new-onset atrial fibrillation with a high ventricular rate.

**Figure 5 jcm-11-04089-f005:**
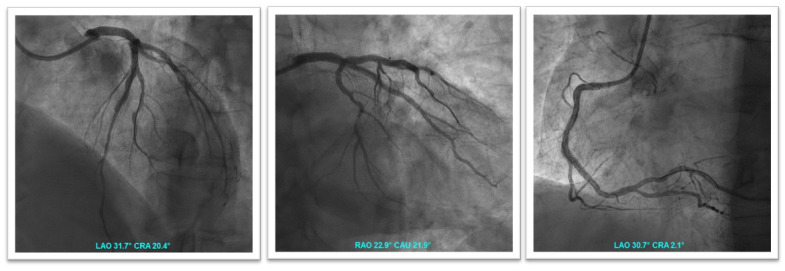
Coronary angiogram revealing the patency of previously implanted stents.

**Figure 6 jcm-11-04089-f006:**
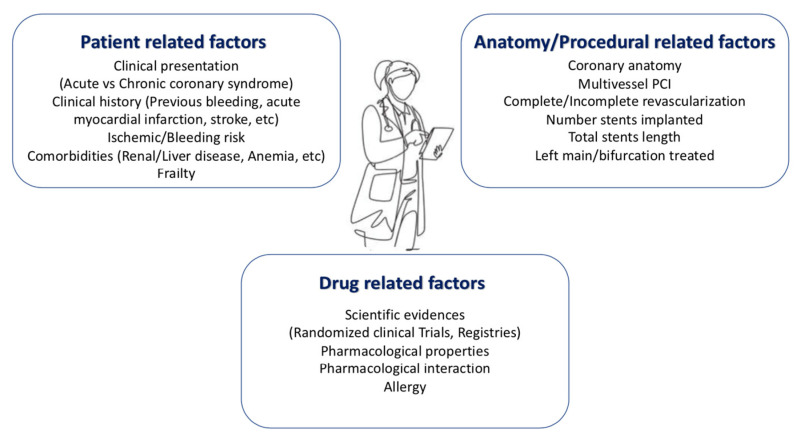
Factors for physicians to consider when determining the optimal antithrombotic regimen for individual patients with atrial fibrillation and acute coronary syndrome or PCI.

**Figure 7 jcm-11-04089-f007:**
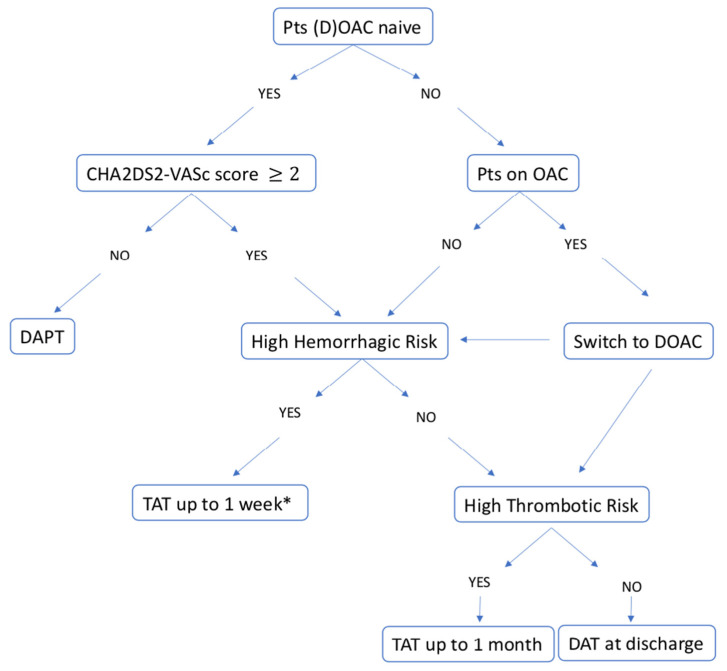
Proposed algorithm for the choice of antithrombotic strategy in patients with atrial fibrillation presenting with acute coronary syndrome and/or undergoing PCI. * evaluate continuation until hospital discharge. DOAC: direct oral anticoagulant; OAC: oral anticoagulant; TAT: triple antithrombotic therapy; DAT: dual antithrombotic therapy.

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
