# Peer review of "A Tailored Antithrombotic Approach for Patients with Atrial Fibrillation Presenting with Acute Coronary Syndrome and/or Undergoing PCI: A Case Series"

_jcm, 2022, doi:10.3390/jcm11144089_

Round 1

Reviewer 1 Report

This is an interesting topic that will be of interest to the readers of the journal. It presents a carefully tailored antithrombotic approaches for five patients with atrial fibrillation presenting with the acute coronary syndrome and/or undergoing PCI. The manuscript is generally well written and structured. It is clear and straight to the point. I have provided few major and revisions below.

Minor revisions:

Line 35: kindly write out the full name before the abbreviation

Figure 4 legend: missing (B)

Line 125: remove dot

Line 140: is it functional mitral regurgitation

Line 146: IVUS kindly provide full name

Line 168: sample instead of simple

Line 173:  missing reference

Care for space bars in lines 165, 173, 174

Lines 180, 181: missing references for the clinical trials

Line 187: missing a point at the end of the phrase

Line 183: missing reference

Line 221: discharged

Line 229: Reference placed before the point

 Line 242: missing reference

Line 245: it would be better to use “irrespective of” instead “irrespectively from”

Major revisions:

The articulation of the decision-making work-up reflects the authors’ full endorsement and implementation of ESC guidelines. Apparently, their tailored antithrombotic approaches are perfect and valid based on the ESC guidelines, however, I think that the manuscript falls short of potential significance, references as well as of some important information with respect to:

£  Full patient history (eg. Absence or presence of hepatic impairment, as this may affect the dose of the selected DOAC)

£  Risk indices for bleeding and ischemic stroke (though presented for some patients). Presenting these values would better reflect the value of the tailored intervention and which direction would it be most helpful, i.e. reducing bleeding or thrombosis

£  Clinical and pharmacological properties of the antithrombotics. For example, the authors could have corroborated their preference of edoxaban over other DOAC in patient 5 based on the fact that no contraindication is applicable for edoxaban in anemic patients.

Although the authors highlighted that none of the 4 RCTs on DOACs and PCI (AUGUSTUS, the PIONEER-AF PCI, the RE-DUAL PCI, and the ENTRUST-AF PCI trial) was powered to address safety concerns with respect to MI-related ischemic events, and accordingly they tended to recommend a personalized strategy with careful consideration of the patient’s ischemic and bleeding risk, yet they have not reported on the follow-up outcomes in terms of benefit on ischemic risk.

Author Response

REVIEWER 1

This is an interesting topic that will be of interest to the readers of the journal. It presents carefully tailored antithrombotic approaches for five patients with atrial fibrillation presenting with the acute coronary syndrome and/or undergoing PCI. The manuscript is generally well written and structured. It is clear and straight to the point. I have provided few major and revisions below:

ANSWER:We are grateful to the reviewer. We followed his/her useful comments which gave us the opportunity to deepen our knowledge about this topic and to improve the paper.
In the revised paper suggested changes and additions have been reported in red.

Minor revision

Line 35: kindly write out the full name before the abbreviation 

Figure 4 legend: missing (B) 

Line 125: remove dot 

Line 140: is it functional mitral regurgitation 

Line 146: IVUS kindly provide full name 

Line 168: sample instead of simple 

Line 173: missing reference 

Care for space bars in lines 165, 173, 174 

Lines 180, 181: missing references for the clinical trials 

Line 187: missing a point at the end of the phrase 

Line 183: missing reference 

Line 221: discharged 

Line 229: Reference placed before the point 

Line 242: missing reference 

Line 245: it would be better to use “irrespective of” instead “irrespectively from

  • ANSWER: We agree with the reviewer.
  • CHANGES: Amended as requested
  •  

Major revision

  1. Full patient history (eg. Absence or presence of hepatic impairment, as this may affect the dose of the selected DOAC)

ANSWER: We are grateful to the reviewer. The clinical history of each patient was reported more comprehensively as required

CHANGES: p. 2 line 71; p. 3, line 96; p. 5, line 120; p 6, line 149; p 7, line 166

  1. Risk indices for bleeding and ischemic stroke (though presented for some patients). Presenting these values would better reflect the value of the tailored intervention and which direction would it be most helpful, i.e. reducing bleeding or thrombosis

ANSWER: We agree with the reviewer. CHA2DS2-VASc and HAS-BLED scores were reported for each patient, as requested

CHANGES: p. 2, line 73; p. 4, line 109; p. 6 line 130; p 6, line 154-155; p 7, line 166-167

3-Clinical and pharmacological properties of the antithrombotics. For example, the authors could have corroborated their preference of edoxaban over other DOAC in patient 5 based on the fact that no contraindication is applicable for edoxaban in anemic patients.

ANSWER: We agree with the reviewer. Recent studies have shown that in patients with atrial fibrillation and anemia, NOAC was associated with lower bleeding risks than warfarin, with no difference in the risk of ischemic stroke/systemic embolism or death (Wang C. et al. Safety and Effectiveness of Non-Vitamin K Antagonist Oral Anticoagulants for Stroke Prevention in Patients With Atrial Fibrillation and Anemia: A Retrospective Cohort Study J Am Heart Assoc. 2019 May 7;8(9):e012029).In particular, in patient 5 we decided to continue with edoxaban which was the DOAC already chosen for this patient due to its clinical characteristics (age, frailty, comorbidity) (Wilkinson C. et al. Clinical outcomes in patients with atrial fibrillation and frailty: insights from the ENGAGE AF-TIMI 48 trial. BMC Med. 2020 Dec 24;18(1):401.)

  1. Although the authors highlighted that none of the 4 RCTs on DOACs and PCI (AUGUSTUS, the PIONEER-AF PCI, the RE-DUAL PCI, and the ENTRUST-AF PCI trial) was powered to address safety concerns with respect to MI-related ischemic events, and accordingly they tended to recommend a personalized strategy with careful consideration of the patient’s ischemic and bleeding risk, yet they have not reported on the follow-up outcomes in terms of benefit on ischemic risk.

ANSWER: ANSWER: We are grateful to the reviewer. The follow-up of each patient is now reported in the revised article

CHANGES: p. 2 line 79-80; p. 4 line 111-112; p 6, line 138-141; p 7, line 171-174

Reviewer 2 Report

Simona Giubilato and colleagues showed that a tailored antithrombotic approach for patients with AF and CAD undergoing PCI as a case series.

In particular, the depictions of the five different processes were very interesting and helpful in clinical setting.

 However, I didn't feel the novelty enough to make a paper.

Author Response

REVIEWER 2

Simona Giubilato and colleagues showed that a tailored antithrombotic approach for patients with AF and CAD undergoing PCI as a case series.

In particular, the depictions of the five different processes were very interesting and helpful in clinical setting.

However, I didn't feel the novelty enough to make a paper ·

ANSWER: We appreciate the comment of the Reviewer and we acknowledge that our article may lack of some novelty. However, our paper sought to provide a practical approach to implement current guidelines and evidence base data into real world clinical practice by reporting a series of paradigmatic clinical cases. We believe that these cases may be of interest to the readers and may serve as pragmatic examples of how to ischemic and bleeding risks are balanced by clinicians when they manage patients with ACS and AF. For this purpose we have added two new figures in the revised work (figure 6 and figure 7). In figure 6 we illustrate the factors to be considered in clinical practice for the choice of the optimal antithrombotic therapy in patients with AF and ACS or PCI. Moreover, in figure 7 we proposed an algorithm for the choice of antithrombotic strategy in this clinical setting.

Reviewer 3 Report

The authors demonstrated  and wildly discussed the case series in the setting of AF and ACS or PCI. The most interesting finding and conclusion is that a patient-tailored approach is crucial for the management of antithrombotic therapy in the setting of AF and ACS or PCI.

Congratulations for the paper concept and presentation of this  important clinical problem.

Author Response

We would like to thank this reviewer for her/his nice comments.

Round 2

Reviewer 1 Report

I would like to thank the authors for addressing my initial comments. The authors have sufficiently improved their paper, in reaction to the comments made. They have provided a nicely detailed and thorough response to the comments from the previous review. They have even improved the introductory part of the discussion. However, the manuscript still requires minor text editing as detailed below:

Figures 6 and 7 are missing

Line 27: should be dual antiplatelet therapy

Line 50: use dual instead of double

Line 53: remove full name as the acronym is defined in line 50

Line 78: kindly write out the full name before the abbreviation

Line 138: remove full name for DAT as the acronym is defined in line 78

Line 192: better to join with previous paragraph

Line 211 and line 212: better to use combined rather than associated

Author Response

We agree with the reviewer. 

CHANGES: Amended as requested.

We upload figures of the paper.

Reviewer 2 Report

Thank you for your revision. I feel the manuscript was well improved.

However, I could not find the newly added figure 6-7. I believe that figure 6-7 have important messages in this paper.

Author Response

We thank the reviewer and apologize that figures 6 and 7 were added in a separate file. We send all the figures of the paper.
